# Analysis of the Batch Effect Due to Sequencing Center in Population Statistics Quantifying Rare Events in the 1000 Genomes Project

**DOI:** 10.3390/genes13010044

**Published:** 2021-12-24

**Authors:** Iago Maceda, Oscar Lao

**Affiliations:** 1Population Genomics, CNAG-CRG, Centre for Genomic Regulation, 08028 Barcelona, Spain; iago.maceda@cnag.crg.eu; 2Barcelona Institute of Science and Technology (BIST), 08036 Barcelona, Spain; 3Universitat Pompeu Fabra (UPF), 08003 Barcelona, Spain

**Keywords:** 1000 Genomes Project, population genetics, batch effect, sequencing center

## Abstract

The 1000 Genomes Project (1000G) is one of the most popular whole genome sequencing datasets used in different genomics fields and has boosting our knowledge in medical and population genomics, among other fields. Recent studies have reported the presence of ghost mutation signals in the 1000G. Furthermore, studies have shown that these mutations can influence the outcomes of follow-up studies based on the genetic variation of 1000G, such as single nucleotide variants (SNV) imputation. While the overall effect of these ghost mutations can be considered negligible for common genetic variants in many populations, the potential bias remains unclear when studying low frequency genetic variants in the population. In this study, we analyze the effect of the sequencing center in predicted loss of function (LoF) alleles, the number of singletons, and the patterns of archaic introgression in the 1000G. Our results support previous studies showing that the sequencing center is associated with LoF and singletons independent of the population that is considered. Furthermore, we observed that patterns of archaic introgression were distorted for some populations depending on the sequencing center. When analyzing the frequency of SNPs showing extreme patterns of genotype differentiation among centers for CEU, YRI, CHB, and JPT, we observed that the magnitude of the sequencing batch effect was stronger at MAF < 0.2 and showed different profiles between CHB and the other populations. All these results suggest that data from 1000G must be interpreted with caution when considering statistics using variants at low frequency.

## 1. Introduction

The 1000 Genomes Project (1000G) [1] corresponds to the first attempt to characterize the worldwide genetic variation in humans. The project was created to generate accurate haplotype information across different human populations. To do so, the project aimed to characterize over 95% of variants in genomic regions that have >1% allele frequency in each of the major population groups (populations in or with ancestry from Europe, East Asia, South Asia, West Africa, and the Americas). The project started with 15 populations in the pilot phase, and had a total of 26 populations by the end of Phase 3, when the project was concluded. In the pilot phase, the project characterized a total of 1092 samples (not evenly distributed across the different populations) and, by the end of the Phase 3, it included a total of 2504 samples (with a close to even distribution across all populations). The final genomic dataset is heterogeneous in nature, comprising individuals with low coverage and exome sequencing data, produced in nine sequencing centers using five sequencing technologies and bioinformatic pipelines. An additional problem is that populations were not divided evenly across all sequencing centers. For example, for the GWD (Gambian in Western Division, Mandinka) population, the BGI sequenced 25 individuals, the Broad Institute sequenced 86 individuals, and Washington University sequenced two individuals. Furthermore, not all centers did all the types of sequencing technologies. For example, the Broad Institute ran low coverage whole genome sequencing and whole exome sequencing. In contrast, the Max Planck Institute for Molecular Genetics only conducted low coverage whole genome sequencing. Furthermore, each center followed its own set of protocols to prepare the samples.

1000G has been one of the cornerstones of population genetics, as it provided the first dataset that considered human worldwide variation. Since then, it has been one of the most widely used human population genetic, medical genetics, and genetic epidemiology datasets. In human population genetics, the 1000G has been widely used for understanding mutation patterns [2], characterizing the genetic variation of the considered human populations [3], identifying segments of archaic introgression [4], studying signatures of positive selection [5,6] or as a gold standard for comparing the burden of loss of function variants (LoF) [3]. In genetic epidemiology, 1000G is routinely used for data phasing and imputation [7,8] to increase single nucleotide variant (SNV) density panels genotyped with microarrays. The prevalence of loss of function (LoF) variants in healthy individuals [9], insights into cancer genomics [10], short tandem repeats variation [11], evolution and functional impact of short indels [12], and many of the results from this project have boosted our understanding of the genomics of the species and the general patterns of diversity present in human populations. Its usage in combination with other whole genome sequencing (WGS) datasets as a reference dataset is also a common practice in the field of medical genomics [3].

However, some problems related to the genotypes called in 1000G have been already reported. For example, Ref. [13] could not reproduce a particular mutation signature (*AC →*CC) reported in the 1000G JPT population (Japanese in Tokyo, Japan) by [14] using a different cohort from the same population (the Nagahama cohort). Furthermore, Ref. [15] evaluated the accuracy of the phasing in the Phase 3 samples and concluded that the 1000G data are best used to impute common variants (minor allele frequency (MAF) ≥0.01) and has limited utility for imputing rare variants. Finally, Ref. [16] described sets of SNVs showing patterns of linkage disequilibrium likely due to the presence of sequencing errors, and directly linked them to the sequencing center where the individual was sequenced.

Singletons can be artificially generated in a variety of ways: the sequencer can misread the base that is being incorporated, the base-calling algorithm may not call the proper base, the alignment algorithm may match the read to an incorrect place, or the SNP-calling or the genotyping algorithm may call an SNV where there is none (or vice versa). In order to solve these problems, a series of different solutions have been proposed such as filtering SNVs by means of a quality score (an associated measure of uncertainty to the SNV) [17], or examining allele distributions across individuals and calculating their fit to an expected distribution [18].

In this study, we analyzed to what extent these batch effects due to the sequencing center could affect measures of genetic variation related to variants at low frequency, looking at rare events that have been previously used in estimating LoF mutations [2], patterns of singletons, and archaic introgression. LoF and particularly dominant mutations are associated with Mendelian diseases [19]. Due to their deleterious nature, they are maintained in general population at a low frequency, except when occurring on dispensable genes or under positive selection [20]. Therefore, being able to distinguish true loss of function mutations from sequencing artefacts is important for properly assessing the genetic risk of an individual regarding a possible medical condition [2]. Singletons are the most common genetic variants in the human genome, reflecting the recent demographic history of human populations [21]. The presence of new mutations can be used for estimating the mutation rate and the environmental and intrinsic factors shaping it [22]. Finally, a depletion of archaic introgression in the human genome, particularly in functional regions, has been documented. This observation is most likely due to purifying selection against the hybrid [23]. However, currently introgressed fragments can have a phenotypic impact (e.g., COVID-19 susceptibility [24]). Therefore, identifying these fragments can be important for understanding the history of archaic introgression, identifying archaic species from which ancient DNA is not yet available [4,25], assessing their role in phenotypes of medical interest.

Furthermore, we wanted to assess to what extent sequencing errors in the 1000G could affect the interpretation of results in newly generated data, especially for populations that due to their rural condition could be potential isolates. As a proof of concept, we used the rural population of the Spanish Eastern Pyrenees (SEP), presented and analyzed in [26].

## 2. Materials and Methods

### 2.1. Datasets

#### 2.1.1. The 1000 Genomes Project

To generate the dataset that we used across this work we downloaded the ready-to-use variant calling format (VCF) files from the FTP site of the 1000 Genomes Project (link). These VCF files are divided by chromosome. We concatenated all the files into a single file using *bcftools* [27]. After this step, we selected those variants that correspond to SNV, and are biallelic and polymorphic across the whole dataset.

We obtained the sequencing center information from the spreadsheet available in the 1000 Genomes Project site (https://www.internationalgenome.org/data/ accessed on 26 April 2019). From this spreadsheet, we selected the Exome sequencing center as our sequencing center reference for all the samples. This decision was made because, for the low coverage whole genome sequencing, some of the samples seem to be sequenced in two different centers according to the spreadsheet (see Supplementary Materials of the 1000 Genomes Phase 3 paper [1] for details). Based on this parameter, we also had to exclude one of the samples from the ACB (African Caribbean in Barbados) population (HG02537).

A summary of the acronyms for sequencing centers and populations used through this article can be found in Appendix A.

#### 2.1.2. Southern Eastern Pyrenees

The SEP dataset consists of 29 samples of unrelated individuals from five different regions of the Spanish Eastern Pyrenees. The characteristics of these samples are described in [26], which studied the basic demographic and spatial aspects of the considered region. Briefly speaking, for each sample it was ascertained that the parents and grandparents were from the same geographic location; samples were healthy with an average age of 76 years and equal proportions of both sexes.

Each sample was sequenced using standard Illumina paired-ends with a read length of 150 bp for an average sequencing coverage of ~40×. The dataset used contains a total of 9,309,056 SNVs.

#### 2.1.3. SNP-Calling of IBS, FIN, and SEP Samples

The bam files from the FIN and IBS samples were downloaded from the 1000 Genomes Project Data Portal site. Samples were SNP-called using GATK HaplotypeCaller v3.6 [28], using the default settings according to the GATK Handbook v3.6 [29] with hs37d5 as the reference assembly. To produce the final VCF file, all the samples (IBS, FIN and SEP) were called jointly.

This step was done to avoid a possible batch effect regarding the different SNP calling methodologies between 1000G and SEP datasets.

#### 2.1.4. Quantification of LoF Variants

To predict loss of function variants, we annotated them using the already annotated table of variants from dbNSFP v2.9.2a [30]. We used SnpSift v4.3e [31] to merge the contents of the table with the VCF file and SnpEff v4.3e [32] to add the functional prediction. From this annotated and functionally predicted VCF file, we decided to use three different algorithms to categorize a variant as LoF: Polyphen2 [33], MutationAssessor [34], and SIFT [31]. We used a conservative approach to call a variant as LoF by only considering LoF variants that were classified as LoF by the three algorithms.

Furthermore, we restricted the analyses to LoF variants that do not appear in homozygosis among samples, as they would reflect redundant or advantageous effects of dispensable genes [20].

#### 2.1.5. Quantification of Derived Singletons

From the general VCF file we extracted singleton SNVs, using the flag AC (number of alternative alleles for that variant across samples) from the INFO field. We used the ancestral allele already present in the dataset (flag AA from the INFO field), which uses the six-way Enredo-Pecan-Ortheus (EPO) primate [35] available in Ensembl v71 [36].

We compared the reference allele to the ancestral allele. If they were the same allele, we marked the singleton as derived. We excluded those SNVs that either had no alignment for the ancestral allele (AA = .), that were considered as a lineage-specific insertion (AA = -), or those in which the allele was not present (AA = N).

#### 2.1.6. Quantification of Archaic Introgressed Alleles

To count the number of introgressed alleles per population, we downloaded the output from the Sprime algorithm used in Browning et al. [4] in 1000G samples. The output from Sprime is divided on population and chromosome basis. These files provide the introgressed alleles for SNVs and population according to the algorithm. We used this information to count the number of introgressed alleles of each individual.

### 2.2. Analyses

#### 2.2.1. Quantification of Sampling Bias between Sequencing Center and Sequenced Populations

To understand whether there was a bias in the portion of samples that each center analyzed from each population, we generated a contingency table (Appendix A) between the populations present in the 1000G and the sequencing center where the exome was generated. With this table, we ran a correspondence analysis using the function *CA* from the R package *FactoMineR* [37].

#### 2.2.2. Quantification of Batch Effect

In order to quantify the batch effect of the sequencing center in the studied statistics of human population genetics for the populations present in 1000G, we used the R package *lme4* [38] to generate hierarchical mixed models controlling for the random effects of the continent and population of assignation of each individual of the type:lmer(log(S)~ SeqCeter+(1|Continent/Pop)
where *S* is the variable of interest. The contrast of hypotheses with the mixed null model
lmer(log(S)~(1|Continent/Pop)
was conducted with the analysis of variance (ANOVA) command of R [39].

For this analysis, we only considered sequencing centers that sequenced samples across all populations.

We also analyzed the variants showing an excess of genetic differentiation due to the sequencing centers in YRI, CEU, JPT, and CHB populations. These populations were the first considered in the Phase 1 of the 1000G project [40] and they have been widely used in population genomics.

For each population and SNV, we assigned each individual from that population to each sequencing center and estimated Weir and Cockerham’s Fst [41] among the different sequencing centers. In order to avoid sampling biases, we selected the same number of individuals (*n* = 5) by center, corresponding to the minimum number of sequenced individuals in CHB at the WUSGC sequencing center. We estimated the $MAF_{P}$ at the population *P* over all the *K* sequencing centers, where $a_{c}$ is the frequency of allele *a* at center *c*:$$P_{a}^{p}=\frac{\sum_{c=1}^{K}a_{c}}{K\times n}$$
$$MAF_{P}=min\left(P_{a}^{p},1-P_{a}^{p}\right)$$

Furthermore, since it has been shown that the magnitude of Fst is dependent on the MAF [42], for each population we estimated the maximum Fst differentiation that could be obtained between centers with the observed $MAF_{P}$ for each SNV. The maximum genetic differentiation is obtained when the first *K* sub-populations (i.e., sequencing centers) share the $K*n*MAF_{P}$ alleles that defines the $MAF_{P}$, and the remaining 4-K sub-populations carry the other allele. We used this maximum observable Fst value given an MAF to scale the observed Fst value for each SNV. This allowed us to compare Fst values with different MAFs.

Next, for all the SNVs with the same MAF, we computed the average Fst and estimated the correlation between Fst and MAF by means of Spearman’s correlation. In order to compute the expected Fst in the absence of sequencing center batch effects, for each population we randomly distributed the individuals among centers, computed the Fst values for each SNV, and computed the mean of each MAF category. We fitted a linear model between the observed and the mean of Monte Carlo-generated Fst values to quantify the departure from the expected Fst under no sequencing center bias.

## 3. Results

### 3.1. Bias in the Samples Sequenced by Sequencing Center

We first checked whether each center had sequenced the same proportion of samples from each population. We observed a statistically significant association (χ^2^ *p*-value = 7.381 × 10^−146^) between certain populations and sequencing centers, as shown by the correspondence analysis on Figure 1. Some populations tend to be overrepresented at certain sequencing centers, even at the continental level. In principle, this implies that any batch effect differentially affecting a sequencing center is going to be reflected by spurious increase of population differentiation and higher than expected heterogeneity within each continent.

### 3.2. Dependence of LoF with the Sequencing Center

Next, to understand the putative effect of batch effects on statistics focusing on rare events, we studied the number of LoF alleles in each of the 1000G individuals. Assessing the biological impact of in silico-predicted LoF is usually complex [43]. Therefore, we adopted a conservative approach for predicting recessive damaging variants that severely affect the function of protein-coding genes. We used variants consistently predicted as highly deleterious by Polyphen2 [32], MutationAssessor [34], and SIFT [31] algorithms. We further restricted our analyses to LoF variants that do not appear in homozygosis [20]. By doing such filtering, we created a putative bias among populations (i.e., populations that are more genetically diverse will tend to have more chances to have homozygote pseudo-LoF genotypes and therefore be removed from the analyses). However, this should not influence the comparisons of centers within each population (i.e., see Figure 2). We ran a hierarchical mixed model in which the dependent variable was the *log*(*LoF*) per individual, the fixed variable was the sequencing center (BGI, BI and WUGSC), and the random effects were nested by continent and population. We observed statistically significant differences between a mixed model that includes the sequencing center as a variable (ANOVA *p*-value = 4.87 × 10^−20^) (Table 1). Next, we wondered if we would observe such batch effect bias in mutations in coding regions classified by the three LoF predictors as benign (Figure 3). In this case, the hierarchical mixed model also supports the role of the sequencing center (ANOVA *p*-value =8.315 × 10^−8^) (Table 2). As we are consider LoF, and set the threshold to absence of homozygotes, any sequencing error occurring in a gene will likely tend to produce a false positive that will be recovered by the three algorithms [9]. Therefore, it is not surprising that the statistical significance was larger in the case of LoF compared to benign, as well as in the magnitude of the estimated slopes (Table 1 and Table 2).

Taken together, the hierarchical mixed models of the LoF and benign variables suggest that the sequencing center plays an important role as a batch effect. This result agrees with [16], which reported an enrichment of mutation artifacts in genes.

### 3.3. Bias in the Number of Reported Singletons by Sequencing Center

We wondered whether such bias could also be found in the presence of derived singletons in each individual. Mixed models with the *log*(*derived singletons by individual*) and *log*(*singletons by individual*) also support a role of the sequencing center (ANOVA *p*-value for derived singletons = 9.81 × 10^−10^; ANOVA *p*-value for singletons = 1.28 × 10^−9^). However, in this case not all the sequencing centers equally contribute to the bias (see Table 3 and Table 4), suggesting some heterogeneity between the centers (see Figure 4 and Figure 5). For example, whereas WUGSC tends to decrease the number of derived singletons observed in the individuals sequenced at that center, BI increases the number of derived singletons and BGI does not significantly affect this variable.

### 3.4. Identification of Introgressed Fragments and Archaic Introgression

Given these results, we studied if the batch effect due to the sequencing center could affect the estimation of variables of population genetics that use the derived alleles to make inferences. One of these variables is the identification of chunks of DNA that are enriched for derived alleles, which under certain demographic models are indicative of the presence of archaic introgression [4]. Furthermore, it has been shown that the presence of archaic introgression depends on the function of the genomic region, as archaic introgression is depleted in genomic regions that contain genes [44,45]. We used the archaic regions from [46] that were identified in the 1000G samples, and computed the number of archaic alleles that are found in each individual. No statistically significant differences were observed between a mixed model using the *log*(*number of introgressed alleles*) and the sequencing center and one without the sequencing center (ANOVA *p*-value = 0.88; Table 5; see Figure 6), thus suggesting that these regions are not enriched by batch effect artifacts due to sequencing center. Nevertheless, since we are considering the estimated amount of archaic introgression per individual given the described introgressed alleles rather than the length of the introgressed chunks of each individual, our results must be considered as quite conservative.

### 3.5. The Pattern of Sequence-Based Outlier SNVs per Population

Next, we studied the properties of genetic markers showing strong deviations between sequencing centers for CEU, CHB, JPT, and YRI. For each population and SNV, we computed Weir and Cockerham’s Fst [41] using the sequencing centers as groups, scaling each Fst value by the maximum Fst value that could be achieved given such MAF, and assigning each SNV to a MAF bin. Our results show that the mean scaled differentiation between sequencing centers of each MAF category depends on the MAF of the SNV independent of which population is considered (Figure 7). In particular, for MAFs close to 0, the amount of genetic differentiation that can be observed between centers is close to the maximum value that can be observed. In contrast, for SNVs with MAFs close to 0.5, the genetic differentiation that is observed between centers is close to 0. For all the populations, a statistically significant Spearman correlation was observed between MAF and the mean of the scaled Fst for the SNVs showing that particular MAF (JPT: rho = −0.914, *p*-value < 10 × 10^−16^; CEU: rho = −0.914, *p*-value < 10 × 10^−16^; CHB: rho = −0.915, *p*-value < 10 × 10^−16^; YRI: rho = −0.914, *p*-value < 10 × 10^−16^).

To test the presence of differences in the sequencing error rate, we generated the null distribution of the scaled Fsts given a MAF under the hypothesis that differences among centers were due to randomness. We observed that the Monte Carlo-scaled Fst showed the same trend as the observed in the real data. Nevertheless, as expected by the lack of differential spurious mutations due to the sequencing center, the simulated Fsts were on average much smaller than the observed ones for each population (slope in a linear model for JPT: 3.932, *p*-value < 2 × 10^−16^; slope CEU: 3.896, *p*-value < 2 × 10^−16^; slope CHB: 3.2, *p*-value < 2 × 10^−16^; slope YRI: 3.52, *p*-value < 2 × 10^−16^).

### 3.6. Effect of the 1000G Bias When Using Other Populations

We decided to test if these batch effects can have a real effect when using the 1000 Genomes Project as a general population dataset when analyzing newly described populations. To test this scenario, we used the dataset presented in [26], which consists of 29 individuals from the Southern Eastern Pyrenees, effectively a rural population.

From the 1000 Genomes Project, we selected the IBS and FIN populations. IBS was selected as it is the most similar population to the SEP dataset found in the 1000 Genomes Project dataset. The FIN population was selected as they are considered a genetic isolate within Europe [47] and to serve as a comparison to rural populations. To reduce the possible batch effect due to the different SNP calling algorithms used in both projects, we performed the same SNP calling in both sets of samples (see Section 2).

Using the combined dataset, we predicted the number of damaging alleles using the same filtering procedure shown above. As shown in Figure 8, there are notable differences between the sequencing centers in both the FIN and IBS populations. These differences were confirmed through a Wilcox test between the different sequencing centers within the FIN and IBS populations (see Table 6 and Table 7 for *p*-values).

## 4. Discussion

The finalization of the 1000G project represented a milestone for the human population genetics community [1]. This dataset is normally used as basis for imputation in microarrays [48], to phase genomes [7] in evolutionary studies [2,4,5,49], in multiple medical studies [50], or as a basis to identify potential genetic isolates [3].

However, two studies [13,15] have recently raised concerns about the presence of batch effects at variants at low frequency. The study [15] in particular pointed to the sequencing center as one of the main factors affecting the presence of rare spurious mutations in 1000G individuals. Given these results, in this study we looked at to what extent the sequencing center, as reported by the spreadsheet of the 1000G (https://www.internationalgenome.org/data/ accessed on 26 April 2019), could influence statistics of population genomics that quantify variants at low frequency in the human genome. Across this paper we have presented proof that some variation present in the 1000 Genomes Project dataset seems to have a noticeable batch effect regarding the sequencing center of the sample, although this batch effect is not noticeable for common variants (MAF > 5%).

LoF variants are usually present at low frequency in the population due to their deleterious effects in carriers and their consequent erasure from the population by purifying selection [51]. Given the fact that 1000G individuals are healthy [1], the presence of LoF variants is expected in heterozygosis and to be mostly recessively inherited. Therefore, the presence of homozygote LoF variants in an individual should either be explained by a relaxation of the purifying selection [52] or by the presence of sequencing errors. In fact, given the evolutionary constraints of LoF, it could be expected that the power for identifying NGS sequencing errors should be enhanced at LoF variants. Sequencing errors typically occur in 0.1–1% of the sequenced nucleotides [53]; as they are rare events, it can be expected that they will tend to produce rare mutations. Several factors shape the probability of a nucleotide to be erroneously sequenced; in particular, poor-quality bases due to low deep sequencing can be misinterpreted by the sequencers [17]. It has been shown that even in the whole exome sequencing of the 1000G, there exists variation in depth of sequencing in the different sequencing centers [54]. Furthermore, there is heterogeneity in the error rates across sequencing platforms [55]. Given that the different populations were non-homogeneously sequenced at the different centers (Figure 1) and that each center used a different technologies or methodologies, differences in the LoF fraction within populations and among genetically related populations due to the sequencing center can be expected. In fact, we have observed that for some populations, the number of LoF variants in a sample is directly dependent on the sequencing center used for the different samples. These results suggest a substantial bias in LoF identification due to sequencing center, even when using stringent filters for defining LoF.

Considering the use of the 1000G for calibrating isolated populations and for comparing the presence of LoF (i.e., see [3]), we wondered to which extent the sequencing center could affect the conclusions regarding the condition of an isolated population. We used the SEP population [26], which has a similar genetic ancestry to the IBS population and which has been sequenced at ~40× coverage. Compared to IBS and FIN, classically considered isolated populations in Europe showing an excess of private Mendelian diseases [47], the SEP population has a lower number of LoF damaging alleles compared to both populations. From a biological point of view, this result could be explained by the presence of inbreeding due to isolation and strong purifying selection acting for a long period in the population (as shown with gorilla populations in [56]). However, when considering the whole genetic variation present in SEP individuals, they genetically resemble IBS individuals and there is no evidence of the presence of long runs of homozygosity or other statistics suggesting long ongoing isolation and inbreeding (see [26]). Moreover, the observed differences between SEP, FIN, and IBS depend on which sequencing center produced the sequences (Table 6 and Table 7). Whereas for BCM the differences in LoF among populations are negligible, for BGI sequenced samples IBS would have an enrichment of LoF variants compared to FIN, which would invalidate previous findings about the population genetics of Finish people [47]. Overall, these results suggest that the observed differences in LoF are more likely due to biases due to the sequencing center. The fact that we observed similar results in the 1000G as those for LoF variants when considering benign mutations suggests that the sequencing bias due to sequencing center could be extended to other mutations occurring at low frequency in these samples. In agreement with this hypothesis, we observed that, for singleton mutations, the sequencing center plays a role, although this signal is not as strong as in the case of LoF variants.

We wondered the limit to which biases in the sequencing center could affect the profile of mutations in a population. Using the originally sampled YRI, CEU, CHB, and JPT populations in Phase 1 of the 1000G, we observed that the bias among centers decreases for all the populations as the frequency of the MAF increases in the considered population. This result could be explained by the variance from a given MAF becoming much higher than the variance due to specific sequencing center error rates, effectively masking their effect. To test this hypothesis, we generated the null distribution of scaled Fsts given a MAF under the hypothesis of no differential sequencing center artifacts. We observed that the Monte Carlo Fst showed the same trend as that observed in the real data.

Overall, these results suggest that the effect we observed in singletons is also observed at higher frequencies. However, when analyzing the detected effect of the sequencing errors on a biological variable that depends on the genetic variation to be detected, such as the level of introgression, we found no effect of the sequencing center on the number of introgressed alleles as defined by [4]. However, it is interesting to note that strong discrepancies were observed in the amount of introgressed alleles for some populations (CHB and CHS; see Figure 6). This is particularly relevant because several studies pointed to the presence of heterogeneous patterns of ghost archaic populations in these populations [4,25].

## 5. Conclusions

In this study, we have shown that statistics that use SNVs that occur at low frequencies in the 1000G are influenced by the sequencing center that generated the samples. Since the proportion of samples generated at each center per population is not the same, conclusions about the biological processes that generated these differences can be jeopardized.

The enrichment of putative biases due to batch effects occur mostly with low frequency variants; in particular, in variants where the sequencing error occurs within a gene, it is more likely that the error will generate a ghost functional error in the protein. Nevertheless, the presence of sequencing errors extends to other MAF categories.

Therefore, when considering such particular kind of statistics and genetic variants, caution must be taken in interpreting the results, particularly when merging with another dataset that may have its own batch effects.

## Figures and Tables

**Figure 1 genes-13-00044-f001:**
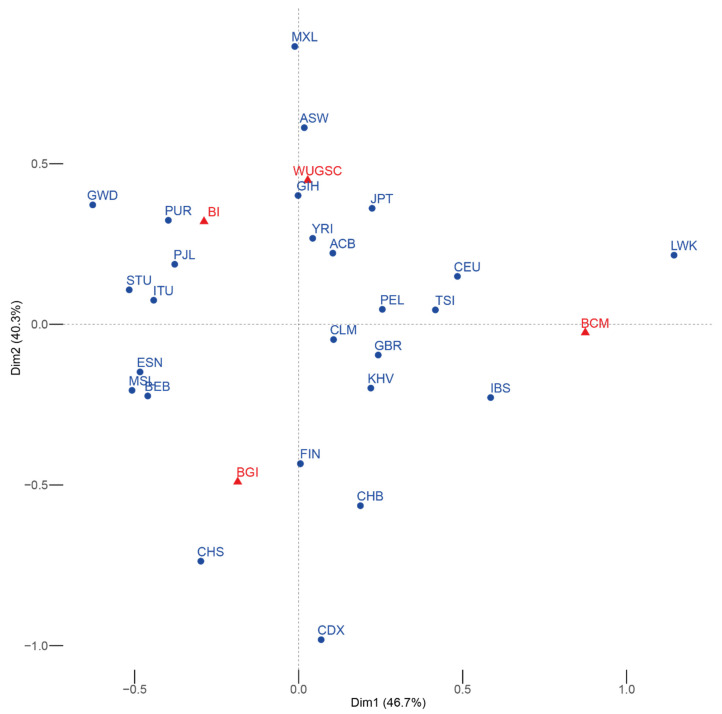
Symmetric correspondence analysis between the 1000G populations (blue) and the sequencing center (red). The expected result of the correspondence analysis in the absence of batch effect is the a group with all the populations roughly equidistant from the different sequencing centers. In this case, some of the populations are closer to a particular sequencing center. Acronyms are showed in Appendix A.

**Figure 2 genes-13-00044-f002:**
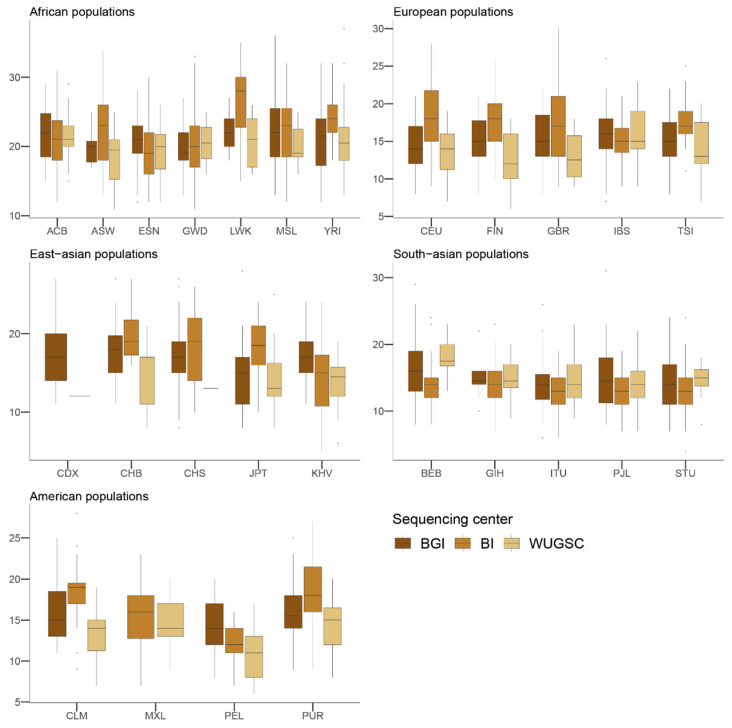
Number of loss of function (LoF) variants by sequencing centers across continental groups and populations. Each panel corresponds to a continental group. In the *x*-axis the name of the population is displayed; in the *y*-axis the number of LoF SNVs per individual is displayed. Acronyms are showed in Appendix A.

**Figure 3 genes-13-00044-f003:**
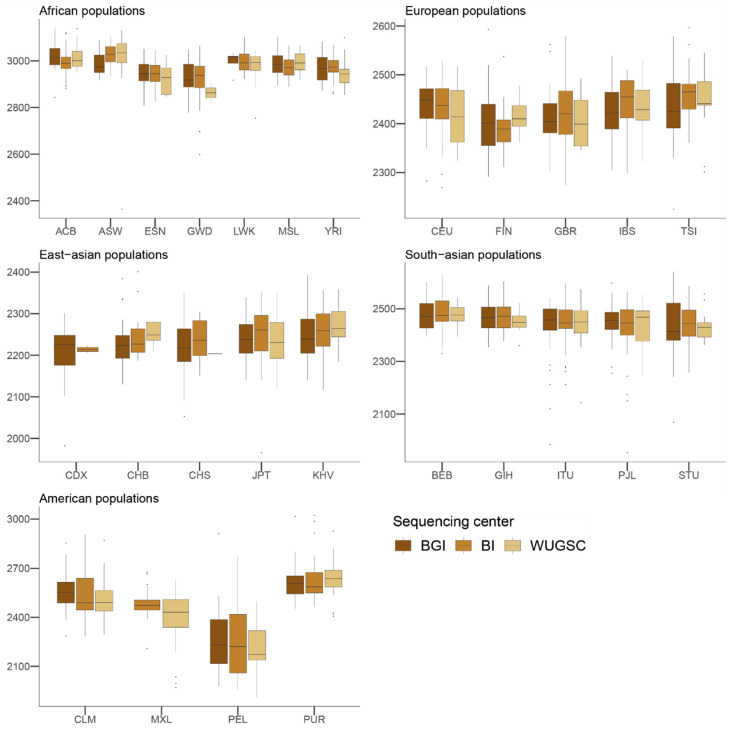
Number of benign variants by sequencing center across continental groups and populations. Each panel corresponds to a continental group. In the *x*-axis the name of the population is displayed; in the *y*-axis the number of benign SNVs per individual is displayed. Acronyms are showed in Appendix A.

**Figure 4 genes-13-00044-f004:**
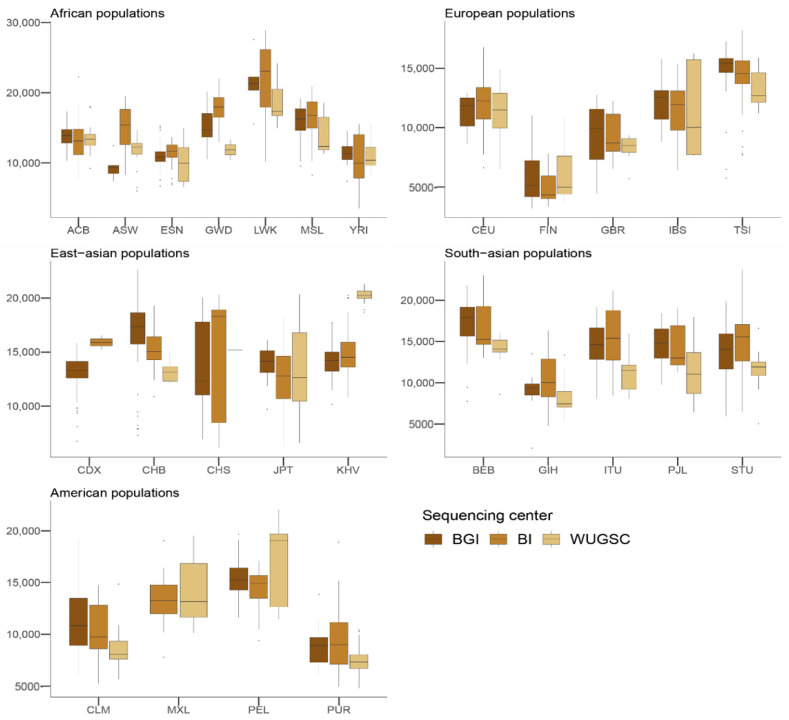
Number of derived singletons by sequencing centers across continental groups and populations. Each panel corresponds to a continental group. In the *x*-axis the name of the population is displayed; in the *y*-axis the number of derived singletons per individual is displayed. Acronyms are showed in Appendix A.

**Figure 5 genes-13-00044-f005:**
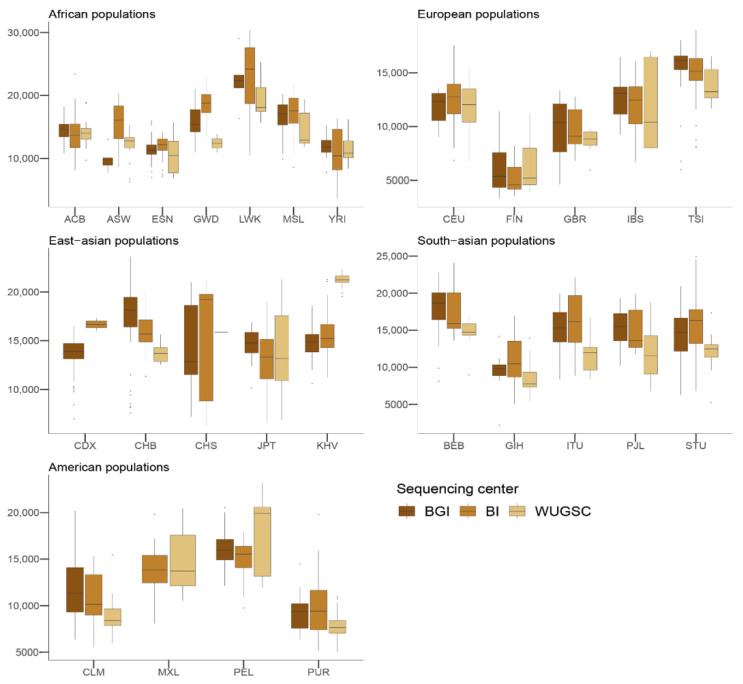
Number of singletons by sequencing centers across continental groups and populations. Each panel corresponds to a continental group. In the *x*-axis the name of the population is displayed; in the *y*-axis the number of singletons per individual is displayed. Acronyms are showed in Appendix A.

**Figure 6 genes-13-00044-f006:**
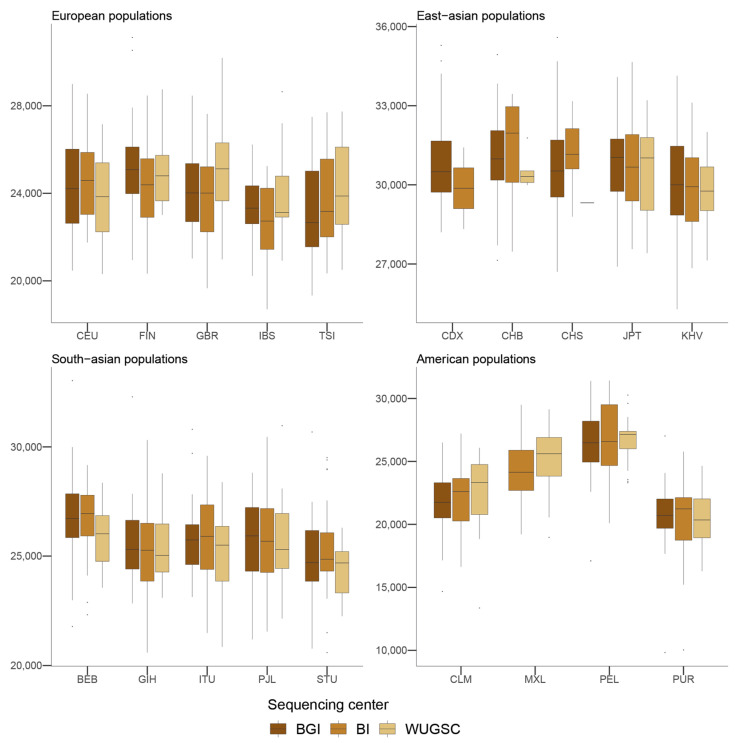
Number of introgressed alleles by sequencing center across continental groups and populations. Each panel corresponds to a continental group. In the *x*-axis the name of the population is displayed; in the *y*-axis the number of introgressed alleles as defined by [4] is displayed. Acronyms are showed in Appendix A.

**Figure 7 genes-13-00044-f007:**
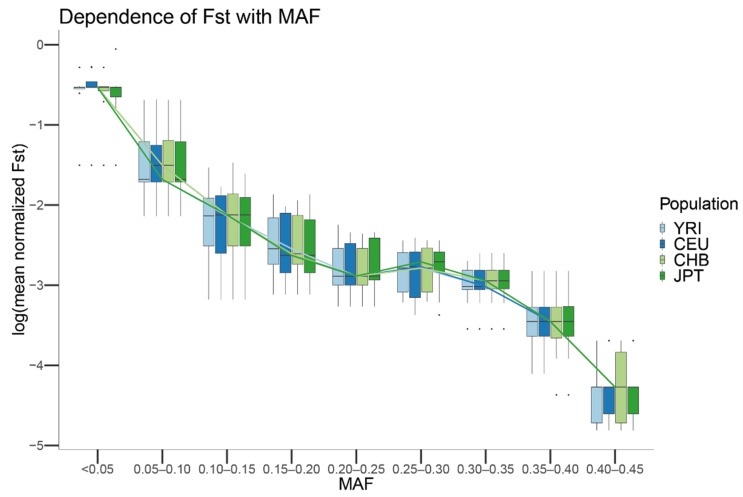
Boxplot of the mean Fst values in log scale by MAF for the CEU, YRI, JPT and CHB populations from 1000G. For each population, the mean Fst of the SNVs with the same MAF was computed. The boxplot was generated by using a MAF binning threshold of 0.05 and plotting the mean Fst in log scale. Lines connect the mean computed for each MAF bin and population.

**Figure 8 genes-13-00044-f008:**
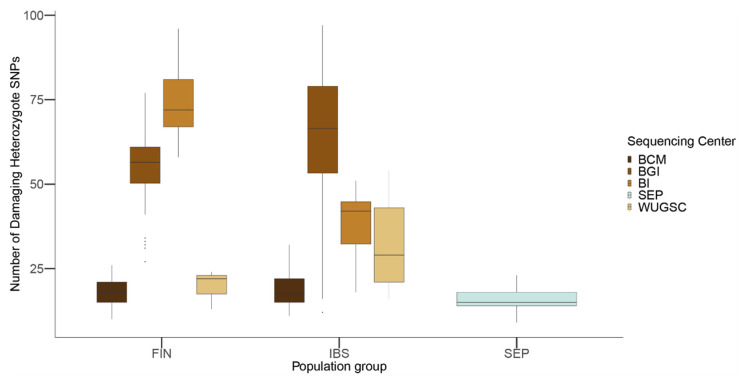
Number of loss of function (LoF) variants by sequencing center in FIN, IBS and SEP populations. In the *x*-axis the name of the population is displayed; in the *y*-axis the number of LoF SNVs per individual is displayed. FIN = 1000G Finland; IBS = 1000G Iberian; SEP = Spanish Eastern Pyrenees.

**Table 1 genes-13-00044-t001:** Coefficient, standard error, and *p* value of the coefficient from the hierarchical mixed model using the number of LoF variants as the dependent variable and the sequencing center as the independent variable. For this analysis, we only considered sequencing centers that sequenced samples across all populations.

Sequencing Center	Coefficient	Std. Error	Pr(>|t|)
(Intercept)	2.660	7.457 × 10^−2^	1.87 × 10^−6^
BGI	1.260 × 10^−1^	1.643 × 10^−2^	2.51 × 10^−14^
BI	1.381 × 10^−1^	1.677 × 10^−2^	2.95 × 10^−16^
WUGSC	4.120 × 10^−2^	1.891 × 10^−2^	0.0294

**Table 2 genes-13-00044-t002:** Coefficient, standard error, and *p* value of the coefficient from the hierarchical mixed model using the number of benign variants as the dependent variable and the sequencing center as the independent variable. For this analysis, we only considered sequencing centers that sequenced samples across all populations.

Sequencing Center	Coefficient	Std. Error	Pr(>|t|)
(Intercept)	7.812	4.735 × 10^−2^	7.21 × 10^−9^
BGI	1.040 × 10^−2^	2.146 × 10^−3^	1.31 × 10^−6^
BI	1.277 × 10^−2^	2.185 × 10^−3^	5.72 × 10^−9^
WUGSC	8.206 × 10^−3^	2.456 × 10^−3^	8 × 10^−4^

**Table 3 genes-13-00044-t003:** Coefficient, standard error, and *p* value of the coefficient from the hierarchical mixed model using the number of derived singletons as the dependent variable. For this analysis, we only considered sequencing centers that sequenced samples across all populations.

Sequencing Center	Coefficients	Std. Error	Pr(>|t|)
(Intercept)	9.413	0.068	5.45 × 10^−9^
BGI	0.024	0.0155	0.1247
BI	0.0328	0.0158	0.0380
WUGSC	−0.0652	0.0177	0.00024

**Table 4 genes-13-00044-t004:** Coefficient, standard error, and *p* value of the coefficient from the hierarchical mixed model using number of singletons as the dependent variable. For this analysis, we only considered sequencing centers that sequenced samples across all populations.

Sequencing Center	Coefficients	Std. Error	Pr(>|t|)
(Intercept)	9.4582	0.0683	5.4 × 10^−9^
BGI	0.0229	0.0156	0.1408
BI	0.0312	0.0159	0.0495
WUGSC	−0.066	0.0178	0.0002

**Table 5 genes-13-00044-t005:** Coefficient, standard error, and *p* value of the coefficient from the hierarchical mixed model using the log(number of inferred archaic introgressed alleles) as defined by [4] and the sequencing center of the samples as the dependent variable. For this analysis, we only considered sequencing centers that sequenced samples across all populations.

Sequencing Center	Coefficients	Std. Error	Pr(>|t|)
(Intercept)	10.15	6.218 × 10^−2^	4.41 × 10^−7^
BGI	1.179 × 10^−3^	6.009 × 10^−3^	0.844
BI	−2.372 × 10^−3^	6.340 × 10^−3^	0.708
WUGSC	1.784 × 10^−3^	7.149 × 10^−3^	0.803

**Table 6 genes-13-00044-t006:** *p*-values from Wilcox tests between the different sequencing centers for the IBS population.

	BGI	BI	WUGSC
BCM	7.789005 × 10^−12^	3.554018 × 10^−7^	1.031651 × 10^−4^
BGI		7.789005 × 10^−12^	3.554018 × 10^−7^
BI			3.554018 × 10^−7^

**Table 7 genes-13-00044-t007:** *p*-values from Wilcox tests between the different sequencing centers for the FIN population.

	BGI	BI	WUGSC
BCM	6.330849 × 10^−10^	6.836831 × 10^−7^	6.825475 × 10^−2^
BGI		6.330849 × 10^−10^	6.836831 × 10^−7^
BI			6.836831 × 10^−7^

## Data Availability

Not applicable.

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
