# Peer review of "Analysis of the Batch Effect Due to Sequencing Center in Population Statistics Quantifying Rare Events in the 1000 Genomes Project"

_genes, 2021, doi:10.3390/genes13010044_

Round 1
Reviewer 1 Report
The manuscript from Maceda and Lao investigate an extremely important question in human population genetics, i.e., the robustness of the 1000G dataset to possible bias due to the different sequencing centers (and technologies) involved in the project. They found strong evidences suggesting how the data produced are not homogeneous and care should be used when interpreting population genetics statistics based on low frequency variants. The manuscript is clear and the analyses sounded: I have just some minor comments, in the hope they could improve this nice work.
The main message of this work is that different sequencing centers provide a different patter of low frequency derived alleles. While the authors focus on the number of LoF and singletons, I would be worried that the phenomenon is more general. I suggest the authors to estimate if possible the SNP density as well as the shape of the normalized SFS (Lapierre et al. 2017, genetics) to check if there is an effect of the sequencing center. Of course, this should also be related, if possible, to coverage.
The materials and methods section it is not well organised. It is not clear the choice of organizing the section into dataset vs analyses, since there is quite some overlapping in what each sub-paragraphs contains. I would rather make more explicit which analysis have been performed on which dataset, since the way it is written hides this basic information.
In the ms the authors only slightly mention about the coverage. Could this variable be responsible of the differences found between the sequencing center? I do not know if there exists already a comprehensive list of how (library preparation, sequencing machine, depth) and where samples where sequenced, but would it be possible to provide this information in the supplementary materials?
Minor issues:
- Title : i am not sure it is a good idea to talk about « quantifying rare events », it is not very precise. I would omit this and let the readers discover where most of the bias seems to be (basically, in statistics measuring/depending on low frequency variants)
- Line 17 : it is not clear what an « infrequent event » is. On the other hand, the main message (if I got it right) is the presence of a bias in the determination of low frequency variants alleles. I would rather mention this point, rather than letting the reader guess what an infrequent event is.
- Line 20 : « systematically associated with the estimation of … »
- Line 24: I think is “genetic” not “genotypic”. You could also delete the word.
- Line 26: “as well…”, it is not very clear.
- Line 28: again, rather change infrequent events with something indicating that the bias is in the estimation of low frequency variants.
- Line 35-36: The sentence is not clear, it seems there is something missing.
- Line 46: (and in many other instances): please provide the definition of the acronym. It could be good to provide a small note with the list of all acronyms used in the main text, it would make the ms more readable.
- Lines 48-49: Sequencing technologies
- Line 62: Insights?
- Line 85: please avoid using the verb “wondering” throughout the paper, you are actually investigating.
- Line 86: I do not understand “recapitulating”
- Line 94: not clear, singletons are genetic variants, not source of genetic variants. Maybe they are the most spread type of genetic variants?
- Line 95: I would rather delete this sentence, singletons are not necessarily recent mutations and I do not agree that alone can provide an estimation of the mutation rate.
- Line 98: depleted with respect to something (ie. Genomic average vs. Functional regions). Just rewrite the sentence.
- Line 105: to which extent.
- Line 106-107: the sentence sound weird. Maybe you just need to write “rural conditions”, cancelling “population” in between.
- Line 116: please define SNV. Note also that you alternatively use SNV and SNP: please just choose one of the two.
- Line 155-157: please reframe it, not so clear
- Line 162: define what the 6Way EPO alignment is.
- Line 162-163: this is not clear. Once you have defined how you identify the ancestral alleles, does it matter which allele the reference has? I do not understand.
- Line 165: not sure I understand, if one of the two allele is not present, then this is not an SNP. I probably did not understand.
- Line 194-206: it is not clear what procedure was followed. For example, in the CEU population, you computed Fst between CEU individuals sequenced by BI et CEU individuals sequenced by BGI? Or rather the Fst is between real populations, but you have different Fst, one for each sequencing center. Also, I do not understand the procedure to evaluate the significance of these Fst. What does it mean that the first K-subpop has the MAF alleles and the others do not have it? Is the allele fixed in a group of populations?
- Lines 233-235: I do not see why it should be the case, especially if there is purifying selection. I would rather delete this sentence
- Lines 289-290. Please remove “usually at low frequencies”, it does not mean anything in my opinion. Moreover, derived alleles are expected to be on average at lower frequencies compared to ancestral alleles, unless selection and specific demographic events are occurring.
- Lines 294: the reasoning seems circular. I think that it should be more interesting to defined introgressed regions in each dataset separately and then see the difference. Contrary to LoF, which are defined on functional basis, introgressed regions are defined on the basis of the excess of derived alleles, which are apparently directly affected by the sequencing center. Of course, it should be established in which samples exactly such regions have been identified: I think that this would result to be very complicated and I fear that these results cannot be easily interpreted.
- Please define how you computed the MAF. Is it in the whole sample considered? As I mentioned above, this part needs to be clarified in the Methods section.
- Line 366 and 396: extent
- Line 368: again, rather focus on low frequency variants rather than ‘infrequent phenomena’
- Lines 402-404: I do not understand why inbreeding should lower the number of Lof, I would say it should be exactly the opposite. And, theoretically, selection should be less efficient in isolates, so this explanation is quite difficult to understand for me.
- Lines 432-434: I would put less emphasis on this point, which is in contrast to what found for lof and singletons, and it could be possible a bias due to the original wrong definition of introgressed regions (due, in turn, to the sequencing center effect too….).
Author Response
Reviewer #1:
The main message of this work is that different sequencing centers provide a different patter of low frequency derived alleles. While the authors focus on the number of LoF and singletons, I would be worried that the phenomenon is more general. I suggest the authors to estimate if possible the SNP density as well as the shape of the normalized SFS (Lapierre et al. 2017, genetics) to check if there is an effect of the sequencing center. Of course, this should also be related, if possible, to coverage.
Reply: In our previous version of the paper, we already analyzed how the sequencing center affects the SFS by means of computing Fst –which is an estimator derived from the SFS- (see Gutenkunst et al 2009) at each population while controlling by the MAF. We showed (Fig. 7) that putative sequencing error biases are more pronounced when the MAF is low (lines 339-341). Regarding the SNV density and coverage, the aim of this study was to identify biases in the public VCFs that are used by many scientists, rather than understanding the ultimate reasons for such biases. Nevertheless, possible explanations for the biases were already described in the introduction.
The materials and methods section it is not well organised. It is not clear the choice of organizing the section into dataset vs analyses, since there is quite some overlapping in what each sub-paragraphs contains. I would rather make more explicit which analysis have been performed on which dataset, since the way it is written hides this basic information.
Reply: we made more clear which analyses are applied to which dataset.
Line 182-183: “In order to quantify the batch effect of the sequencing center in the studied statistics of human population genetics for the populations present in 1000G…”
In the ms the authors only slightly mention about the coverage. Could this variable be responsible of the differences found between the sequencing center? I do not know if there exists already a comprehensive list of how (library preparation, sequencing machine, depth) and where samples where sequenced, but would it be possible to provide this information in the supplementary materials?
Reply: We agree that coverage is a main issue in 1000G. However, coverage is a metric difficult to ascertain from the public VCF, as it is a mix of different technologies. Regarding where and how samples were sequenced, this information can be found in the supplementary material of the 1,000 Genomes Phase 3 paper (A global reference for human genetic variation, Nature 2015). We have included a sentence explaining the origin of the samples and characteristics of their sequencing (line 123).
Line 123-124: “(see supplementary material of the 1,000 Genomes Phase 3 paper, [1], for details)”
Minor issues:
Title: I am not sure it is a good idea to talk about « quantifying rare events », it is not very precise. I would omit this and let the readers discover where most of the bias seems to be (basically, in statistics measuring/depending on low frequency variants)
Reply: In this paper, we have focused on a particular subset of statistics, which are used in population genomics, which are particularly sensible to variants and events (i.e. introgression) at low frequency. We think that removing “quantifying rare events” can be misleading
Line 17: it is not clear what an « infrequent event » is. On the other hand, the main message (if I got it right) is the presence of a bias in the determination of low frequency variants alleles. I would rather mention this point, rather than letting the reader guess what an infrequent event is.
Reply: We have changed it in the manuscript.
Line 17: “Despite the overall effect of these ghost mutations can be considered negligible for common genetic variants in many populations, it remains unclear the potential bias when studying genetic variants associated to low frequency variants in the population.”
Line 20: « systematically associated with the estimation of … »
Reply: We have changed it in the manuscript.
Line 21: “Our results support previous studies showing that the sequencing center is associated with loss of function mutation (LoF) and singletons, independently of which population is considered.”
Line 24: I think is “genetic” not “genotypic”. You could also delete the word.
Reply: We have changed it in the manuscript.
Line 24: “When analyzing the frequency of SNVs showing extreme patterns of differentiation among centers for CEU, YRI, CHB and JPT, we observed that the magnitude of the sequencing batch effect was stronger at MAF < 0.2, and different profiles between CHB and the other populations.”
Line 26: “as well…”, it is not very clear.
Reply: changed in the manuscript.
Line 26: “When analyzing the frequency of SNVs showing extreme patterns of differentiation among centers for CEU, YRI, CHB and JPT, we observed that the magnitude of the sequencing batch effect was stronger at MAF < 0.2, and different profiles between CHB and the other populations.”
Line 28: again, rather change infrequent events with something indicating that the bias is in the estimation of low frequency variants.
Reply: We have changed it in the manuscript. Changed all instances of “infrequent events” to “variants at low frequency”
Line 35-36: The sentence is not clear, it seems there is something missing.
Reply: We have rephrased the sentence.
Lines 24-28: “This project was created to generate accurate haplotype information across different human populations. To do so, the project aimed to characterize over 95% of variants in genomic regions that have >1% allele frequency in each of the major population groups (populations in or with ancestry from Europe, East Asia, South Asia, West Africa, and the Americas).”
Line 46: (and in many other instances): please provide the definition of the acronym. It could be good to provide a small note with the list of all acronyms used in the main text, it would make the ms more readable.
Reply: We have changed it in the manuscript. The acronyms used for the populations and sequencing centres are described in the Supplementary Table 1.
Line 46: “An additional problem is that populations were not divided evenly across all sequencing centers. For example, for the GWD (Gambian in Western Division, Mandinka) population the BGI sequenced 25 individuals, the Broad Institute sequenced 86 individuals and the Washington University sequenced 2 individuals.”
Lines 48-49: Sequencing technologies
Reply: We have changed it in the manuscript
Line 48-49: “Also, not all centers did all the types of sequencing technologies.”
Line 62: Insights?
Reply: We have changed it in the manuscript.
Line 85: please avoid using the verb “wondering” throughout the paper, you are actually investigating.
Reply: We have changed it in the manuscript.
Line 91: “In this study we studied to which extent these batch effects due to the sequencing center could affect measures of genetic variation related to variants at low frequency, looking at rare events that have been previously used in estimating LoF mutations [2], patterns of singletons and archaic introgression. LoF, and particularly dominant mutations, are associated to Mendelian diseases [19].”
Line 86: I do not understand “recapitulating”
Reply: We have changed it in the manuscript.
Line 91: “In this study we studied to which extent these batch effects due to the sequencing center could affect measures of genetic variation related to variants at low frequency, looking at rare events that have been previously used in estimating LoF mutations [2], patterns of singletons and archaic introgression. LoF, and particularly dominant mutations, are associated to Mendelian diseases [19].”
Line 94: not clear, singletons are genetic variants, not source of genetic variants. Maybe they are the most spread type of genetic variants?
Reply: We have changed it in the manuscript.
Line 100: “Singletons are the most common genetic variants in the human genome, reflecting the recent demographic history of human populations [21].”
Line 95: I would rather delete this sentence, singletons are not necessarily recent mutations and I do not agree that alone can provide an estimation of the mutation rate.
Reply: used this sentence in agreement with reference 21.
Line 98: depleted with respect to something (ie. Genomic average vs. Functional regions). Just rewrite the sentence.
Reply: We have rephrased the sentence.
Line 98: “Finally, a depletion of archaic introgression in the human genome, particularly at functional regions, has been documented. This observation is most likely due to purifying selection against the hybrid [23].”
Line 105: to which extent.
Reply: We have changed it in the manuscript.
Line 105: “Furthermore, we wanted to assess to which extent sequencing errors in the 1000G could affect the interpretation of results in newly generated data, …”
Line 106-107: the sentence sound weird. Maybe you just need to write “rural conditions”, cancelling “population” in between.
Reply: the particular condition of the SEP population is that it corresponds to a rural population (as defined in the paper in which it was described originally).
Line 116: please define SNV. Note also that you alternatively use SNV and SNP: please just choose one of the two.
Reply: We have changed it in the manuscript. All instances of SNP have been changed to SNV.
Line 155-157: please reframe it, not so clear
Reply: We have modified the sentence.
Lines 155-157: “Furthermore, we restricted the analyses to LoF variants that do not appear in ho-mozygosis among samples, as they would reflect redundant or advantageous effects of dispensable genes [20].”
Line 162: define what the 6Way EPO alignment is.
Reply: this corresponds to the description of the AA flag (ancestral allele) of the INFO field in the VCF from the 1,000 Genomes Project. We have included a sentence describing it in line 162.
Line 160-163: “We used the ancestral allele already present in the dataset (flag AA from the INFO field), which uses the 6-way Enredo-Pecan-Ortheus (EPO) primate [35] available in Ensembl v71 [35]”
Line 162-163: this is not clear. Once you have defined how you identify the ancestral alleles, does it matter which allele the reference has? I do not understand.
Reply: the ancestral allele shown in the AA flag can be the same as the reference allele (making the alternative allele the derived allele) or the same as the alternative allele (making the alternative allele the ancestral allele). Although this last pattern is not frequent, we found instances of it in the VCF.
We tried to introduce as little changes as possible to the VCF, and used as much info from the VCF as possible.
Line 165: not sure I understand, if one of the two allele is not present, then this is not an SNP. I probably did not understand.
Reply: This part (when the AA flag does not have a clear base) is to clarify the ancestral state assignation in those cases in which we do not have a proper ancestral allele in the VCF. When we do not find the ancestral allele in the VCF file, we decided excluding it from the final dataset instead of adding another confounding factor in the form of a second ancestral assignment.
Line 194-206: it is not clear what procedure was followed. For example, in the CEU population, you computed Fst between CEU individuals sequenced by BI et CEU individuals sequenced by BGI? Or rather the Fst is between real populations, but you have different Fst, one for each sequencing center. Also, I do not understand the procedure to evaluate the significance of these Fst. What does it mean that the first K-subpop has the MAF alleles and the others do not have it? Is the allele fixed in a group of populations?
Reply: We have modified the material and methods section explaining in detail the procedure we have applied.
Summarizing: for each of the four considered populations, we assign each individual to each of the four sequencing centers. Given that the sample size by center and population is not the same, we subsampled each population and sequencing center to the minimum number of individuals observed at WUGSC and CHB (5 individuals).
For each SNV and population, we compute the Fst between sequencing centers. The minimum allele frequency (MAF) of a SNV for a given population is computed without taking into account the sequencing center.
The maximum genetic differentiation is obtained for a for a given number of MAF alleles when the first K sub-populations (i.e. sequencing centers) have the MAF alleles and the remaining 4-K sub-populations carry the other allele.
Lines 199-224: “For each population and SNV, we assigned each individual from that population to each sequencing center and estimated the Weir and Cockerham’s Fst [39] among the different sequencing centers. In order to avoid sampling biases, we selected the same number of individuals (n=5) by center, corresponding to the minimum number of sequenced individuals in CHB at the WUSGC sequencing center. We estimate the at the population P over all the K sequencing centers, where is the frequency of allele a at center c:
Furthermore, since it has been shown that the magnitude of Fst is dependent on the MAF [40], for each population we estimated the maximum Fst differentiation that could be obtained between centers with the observed for each SNV. The maximum genetic differentiation is obtained when the first K sub-populations (i.e. sequencing centers) share the allele that defines the , and the remaining 4-K sub-populations carry the other allele. We used this maximum Fst value given a MAF to scale the observed Fst value for each SNV. This allowed us to compare Fst values from different MAF categories.
Next, for each MAF category we computed the average Fst, and estimated the correlation between Fst and MAF by means of Spearman’s correlation. In order to compute the expected Fst in the absence of sequencing center batch effects, for each population we randomly distributed the individuals among centers, computed the Fst values for each SNV, and computed the mean of each MAF category. We fitted a linear model between the observed and the mean of Monte Carlo-generated Fst values to quantify the departure from the expected Fst under no sequencing center bias.”
Lines 233-235: I do not see why it should be the case, especially if there is purifying selection. I would rather delete this sentence
Reply: Following the suggestions of the reviewer, we removed this sentence.
Lines 289-290. Please remove “usually at low frequencies”, it does not mean anything in my opinion. Moreover, derived alleles are expected to be on average at lower frequencies compared to ancestral alleles, unless selection and specific demographic events are occurring.
Reply: We have modified the sentence.
Line 328: “Given these results, we studied if the batch effect due to the sequencing center could affect the estimation of variables of population genetics that use the derived alleles to make inferences.”
Lines 294: the reasoning seems circular. I think that it should be more interesting to defined introgressed regions in each dataset separately and then see the difference. Contrary to LoF, which are defined on functional basis, introgressed regions are defined on the basis of the excess of derived alleles, which are apparently directly affected by the sequencing center. Of course, it should be established in which samples exactly such regions have been identified: I think that this would result to be very complicated and I fear that these results cannot be easily interpreted.
Please define how you computed the MAF. Is it in the whole sample considered? As I mentioned above, this part needs to be clarified in the Methods section.
Reply: We agree that it would be better to identify the introgressed fragments at individual base. However, as explained in the paper, this information was not provided by the original paper. We have included a sentence explaining that this is a limitation of our study:
Line 322-324: “Nevertheless, since we are considering the estimated amount of archaic introgression per individual given the described introgressed alleles rather than the length of introgressed chunks of each individual, our results must be considered as quite conservative.”
How MAF is estimated is described in material and methods:
Line 210-212: “Furthermore, since it has been shown that the magnitude of Fst is dependent on the minimum allele frequency (MAF) [40], we estimated the maximum Fst differentiation that could be obtained with the observed MAF in each population for each SNV.”
Line 366 and 396: extent
Reply: We changed it in the manuscript.
Line 392-395: “Given these results, in this study we looked to which extent the sequencing center, as reported by the spreadsheet of the 1000G (https://www.internationalgenome.org/data/), could influence statistics of population genomics that quantify variants at low frequency in the human genome.”
Line 421-423: “Considering the use of the 1000G for calibrating isolated populations and for comparing the presence of LoF (i.e. see [3]), we wondered to which extent the sequencing center could affect the conclusions regarding the condition of an isolated population.”
Line 368: again, rather focus on low frequency variants rather than ‘infrequent phenomena’
Reply: We changed it in the manuscript.
Line 392-395: “Given these results, in this study we looked to which extent the sequencing center, as reported by the spreadsheet of the 1000G (https://www.internationalgenome.org/data/), could influence statistics of population genomics that quantify variants at low frequency in the human genome.”
Lines 402-404: I do not understand why inbreeding should lower the number of Lof, I would say it should be exactly the opposite. And, theoretically, selection should be less efficient in isolates, so this explanation is quite difficult to understand for me.
Reply: We agree with the reviewer that this is the case under normal conditions. However, as we explained in the manuscript, the only way we could find to explain our results was the hypothesis presented for the Gorilla’s populations (from the referenced paper). Nevertheless, none of the other conditions supporting such explanation met in the SEP population (such as high RoH or distinctive LD patterns).
Lines 432-434: I would put less emphasis on this point, which is in contrast to what found for lof and singletons, and it could be possible a bias due to the original wrong definition of introgressed regions (due, in turn, to the sequencing center effect too….).
Reply: We have removed this sentence.

Reviewer 2 Report
I have no issues with the data of this manuscript or its analysis. The paper is well written and coherent. It is also important because it discusses all the potential “batch effects” that may make the 1000G project less valuable to some or may suggest those who use it employ a grain of salt. However, I do have some issues with the way figures and tables are labelled and explanations in the text.
1) Table 1 & 2. Please be clear. What is being estimated in these tables. What do larger and smaller values mean? WUGSC has the most divergent estimate. You do not even mention this—even if you discuss this in the Discussion section, it does seem like an important thing to at least point out in the results section.
2) Table 3 & 4. Again be clear. What are you estimating—either put this in the table itself or in the text? Do not make readers search.
3) Tables 5 & Figure 6. I am very confused why these are not mentioned in Section 3.4. Please cite the result in its proper section. It is pretty standard in scientific papers to indicate the appropriate table or figure with its presentation in the results section.
4) I am very confused by the result presented in section 3.4. The estimate may be the same for each model, as indicated, but please address the meaning of the estimate data in Table 5. The three sequencing centers do not have the same estimate here. Again, PLEASE clarify what the estimate is estimating. Estimates in the introgression identified? Something else? It’s very confusing.
5) Please make sure that at some point every acronym is defined. Not everyone reading this paper will be familiar with the jargon.
Author Response
Reviewer #2
1) Table 1 & 2. Please be clear. What is being estimated in these tables. What do larger and smaller values mean? WUGSC has the most divergent estimate. You do not even mention this—even if you discuss this in the Discussion section, it does seem like an important thing to at least point out in the results section.
Reply: We have rephrased the Legend of all tables. We have also included a sentence in the Material and Methods explaining which sequencing centers are used for this analysis.
2) Table 3 & 4. Again be clear. What are you estimating—either put this in the table itself or in the text? Do not make readers search.
Reply: See comment 1.
3) Tables 5 & Figure 6. I am very confused why these are not mentioned in Section 3.4. Please cite the result in its proper section. It is pretty standard in scientific papers to indicate the appropriate table or figure with its presentation in the results section.
Reply: Table 5 was already referred in the results section. We added the reference to Figure 6, next to the reference to Table 5.
4) I am very confused by the result presented in section 3.4. The estimate may be the same for each model, as indicated, but please address the meaning of the estimate data in Table 5. The three sequencing centers do not have the same estimate here. Again, PLEASE clarify what the estimate is estimating. Estimates in the introgression identified? Something else? It’s very confusing.
Reply: We have rephrased the legend to make it clearer. The reported results refer to the slope of the hierarchical linear model considering the sequencing centers while controlling by the random effects due to population and continent.
Line 325-328: “Coefficient, standard error, and p value of the coefficient from the hierarchical mixed model using as dependent variable the log(number of inferred archaic introgressed alleles) as defined by [4] and the sequencing center of the samples. For this analysis, we only considered sequencing centers that sequenced samples across all populations.”
5) Please make sure that at some point every acronym is defined. Not everyone reading this paper will be familiar with the jargon.
Reply: Following the suggestion of the reviewer, we have changed it in the manuscript.
